# Comprehensive Landscape of BRAF Variant Classes, Clonalities, and Co-Mutations in Metastatic Colorectal Cancer Using ctDNA Profiling

**DOI:** 10.3390/cancers16040737

**Published:** 2024-02-09

**Authors:** Benny Johnson, Van Morris, Xuemei Wang, Arvind Dasari, Kanwal Raghav, John Paul Shen, Michael S. Lee, Ryan Huey, Christine Parseghian, Jason Willis, Robert Wolff, Leylah M. Drusbosky, Michael J. Overman, Scott Kopetz

**Affiliations:** 1Department of Gastrointestinal Medical Oncology, Division of Cancer Medicine, The University of Texas MD Anderson Cancer Center, Houston, TX 77030, USA; 2Department of Biostatistics, The University of Texas MD Anderson Cancer Center, Houston, TX 77030, USA; 3Guardant Health, Redwood City, CA 94063, USA

**Keywords:** atypical *BRAF*, metastatic colorectal cancer, *BRAF* mutation, *KRAS* mutation, *NRAS* mutation

## Abstract

**Simple Summary:**

Patients with atypical (nonV600) *BRAF* mutations represent a unique category of metastatic colorectal cancer patients. In the past, these patients were considered to be similar to those with the more common *BRAF^V600E^* mutation. However, additional investigation confirmed patients with atypical *BRAF* mutations have a distinct molecular make up, clinical course, and treatment response to both chemotherapy and targeted therapy, compared to those of patients with *BRAF^V600E^* mutations. Here, we report the key characteristics of patients with atypical *BRAF* mutations identified from a large circulating tumor DNA database and a real-world clinical cohort, highlighting important differences, such as the presence of additional mutations and related survival outcomes. These findings support the need for dedicated research efforts to understand the intricacies of atypical *BRAF* mutations in colon cancer and promote the discovery of new therapies for these patients.

**Abstract:**

Although V600E accounts for the majority of the *BRAF* mutations in metastatic colorectal cancer (mCRC), non-V600 *BRAF* variants have been shown in recent years to represent a distinct molecular subtype. This study provides a comprehensive profile of *BRAF* variants in mCRC using a large genomic database of circulating tumor DNA (ctDNA) and analyzing clinical outcomes in a cohort of patients with atypical (non-V600) *BRAF* variants (*aBRAF*; class II, class III, unclassified). Overall, 1733 out of 14,742 mCRC patients in the ctDNA cohort had at least one *BRAF* variant. Patients with atypical *BRAF* variants tended to be younger and male. In contrast to *BRAF^V600E^*, *BRAF* class II and III variants and their co-occurrence with *KRAS*/*NRAS* mutations were increased at baseline and especially with those patients predicted to have prior anti-EGFR exposure. Our clinical cohort included 38 patients with atypical *BRAF* mCRC treated at a large academic referral center. While there were no survival differences between atypical *BRAF* classes, concurrent *RAS* mutations or liver involvement was associated with poorer prognosis. Notably, patients younger than 50 years of age had extremely poor survival. In these patients, the high-frequency *KRAS*/*NRAS* co-mutation and its correlation with poorer prognosis underlines the urgent need for novel therapeutic strategies. This study represents one of the most comprehensive characterizations to date of atypical *BRAF* variants, utilizing both ctDNA and clinical cohorts.

## 1. Introduction

Colorectal cancer (CRC) is a common malignancy, ranking fourth in cancer diagnosis and second in cancer-related deaths in the US [1,2]. Many patients develop metastatic colorectal cancer (mCRC) due to limitations in early detection as well as significantvariability in clinical presentation. Furthermore, considering the increasing incidence of young onset colorectal cancer the identification of innovative strategies to improve mCRC treatment remains a critical unmet need [2]. Additionally, the consistent success of immune checkpoint blockade to date has been limited to patients with microsatellite instability high (MSI-H)/mismatch repair deficient (dMMR) colorectal cancer, leaving the remaining microsatellite stable (MSS) patients without any novel immunotherapy-based treatment options beyond those administered through clinical trials [3,4,5,6,7,8,9,10,11,12].

*BRAF* mutations represent one of the most common aberrations in human malignancies [13], including mCRC [14]. BRAF is a serine threonine kinase downstream of RAS, a part of the mitogen-activated protein kinase (MAPK) signaling pathway. The most common *BRAF* mutation, present in 7–10% of patients [13,14], occurs at codon 600, with a valine to glutamic acid change (c.1799T>A or p.V600E). This mutation results in the RAS-independent constitutive activation of MAPK, with the promotion of uncontrolled tumor cell proliferation and metastases formation, which lead to poorer outcomes in terms of patient survival [13,14]. *BRAF^V600E^* and *RAS* mutations are predominantly mutually exclusive [3,15,16].

Atypical, non-V600 BRAF (*aBRAF*) mutations have been recognized in recent years as a unique molecular subset, partly due to the increased use of expanded molecular profiling and circulating tumor DNA (ctDNA) analysis in the management of mCRC. Preclinical data specifically characterized *aBRAF* mutations into class II or class III designations, based on their underlying signaling biology [17]. Class II *aBRAF* mutants signal via constitutive dimerization and are RAS-independent [17], while Class III *aBRAF* mutants are characterized by low or absent kinase activity and are associated with RAS activation via receptor tyrosine kinase signaling [17].

Previous studies reported that *aBRAF* mutations are present in approximately 2.2% of patients with mCRC, with an improved median OS of 60.7 months compared to 11.7 months for patients with *BRAF^V600E^* mutations [18]. However, considering the chronicity of *aBRAF* mCRC, and the fact that patients still succumb to the disease, novel treatments are still needed for personalized treatment and to preserve patients’ quality of life. Furthermore, *aBRAF* mCRC has a distinct, antithetical clinical profile from *BRAF^V600E^* mCRC. Most patients present microsatellite stable (MSS) disease, left-sided primary tumors, lower-grade histology, and the clinical presentations of non-peritoneal spread [19,20]. *RAS* co-mutations may also be present [19,20].

To date, there are no specific guidelines for the management of patients with *aBRAF* mCRC. Early-phase clinical trials are recommended after the failure of traditional systemic chemotherapy. In particular, anti-EGFR therapy is known to elicit a poor response in *BRAF^V600E^* patients [21]. The potential utility of EGFR inhibitors in *aBRAF* patients is less clear; however, a recent study showed the response to anti-EGFR treatment to be poor in both class II and class III patients, and that these mutations were significantly more common in patients previously treated with EGFR inhibitors, suggesting that they may represent a novel resistance mechanism [22].

In this study, we aimed to provide a comprehensive landscape of *BRAF* mutations in mCRC, analyzing the distribution of different mutation classes, their clonalities, and the frequency of co-mutation with other genes of interest through ctDNA profiling. Moreover, we studied how these different molecular characteristics correlated with clinical outcomes in our MD Anderson cohort to confirm previous findings, suggesting that the *RAS-aBRAF* co-mutated phenotype may represent a more aggressive subclass of mCRC than those previously described.

## 2. Methods

### 2.1. Patient Population

We classified all *BRAF* mutations in this study based on the published preclinical study as summarized above [17]. The first cohort analyzed for this study included patients with mCRC in the Guardant Health (GH) database from September 2014 to May 2021. These data were used to perform a retrospective review. The Guardant360 targeted sequencing assay is a blood-based, “liquid” biopsy that identifies single nucleotide variants (SNVs), indels, copy number amplifications, and fusions within the protein-coding regions of up to 83 genes. Treatment history was not available for this cohort; therefore, a previously validated and highly specific score was used to predict the anti-EGFR exposure status [22,23]. Briefly, patients were considered anti-EGFR exposed, if the ctDNA analysis revealed the specific molecular abnormalities consistent with a previous exposure [23]. The second cohort (clinical cohort) included 38 patients with *aBRAF* mCRC (all 38 patients were confirmed via tissue next generation sequencing; 10 pts with ctDNA testing) who received treatment at MD Anderson Cancer Center between June 2018–January 2022. These patients were analyzed based on their treatment history and overall survival (OS). Next generation sequencing is a tissue-based assay performed with an in-house panel at MD Anderson Cancer Center, covering an estimated 600 genes, by utilizing patient primary tumor or metastatic site biopsies obtained as standard of care.

### 2.2. Data and Statistical Analysis

*BRAF* amplifications and synonymous variants were not included in this study. A variant was defined as clonal, if the allele frequency (VAF) was greater than 50% of the highest somatic VAF in the sample; otherwise, it was defined as subclonal. When multiple samples were available for a patient, only the earliest tested sample was included in the analysis.

Fisher’s exact test was used in the analysis comparing molecular classes and variant groups. OS was defined as time from mCRC diagnosis to the date of death or last follow-up. Kaplan–Meier survival curves and long-rank tests were used to compare OS between patient groups. Statistical analyses were performed using GraphPad Prism, version 9.3.1 (GraphPad Software, La Jolla, CA, USA). A value of *p* < 0.05 was considered statistically significant.

## 3. Results

### 3.1. Patient Characteristics

The GH cohort consisted of 14,742 patients with mCRC, among whom 1733 presented at least one *BRAF* variant (Table 1). Based on *BRAF* variant classes, there were 926 (6.3%) patients with *BRAF^V600E^* mutations, 159 (1.1%) patients with class II *BRAF* variants, 277 (1.9%) patients with class III *BRAF* variants, and 475 (3.2%) patients with unclassified *BRAF* variants (Table 1). While the cohort with *BRAF^V600E^* mutations had more female (55.9%) and older (≥65 years, 51%) patients, there were more male (56.2–60%) and younger (<65 years, 58.8–62.9%) patients in the groups with *aBRAF* variants (Table 1).

### 3.2. Distribution of BRAF Variant Classes

431 unique *BRAF* variants were identified out of a total of 1905, excluding amplification and synonymous variants. Within the total BRAF variants, 926 (48.6%) were V600 mutations, followed by 163 (8.6%) class II variants, 284 (14.9%) class III variants, and 532 (27.9%) unclassified variants (Table 1). The most frequently mutated codons in class II and III variants were G469 and D594 (Figure 1A,B), respectively. Other commonly mutated codons included K601, G464, and L485 in class II (Figure 1A), as well as N581, G466, and G596 in class III (Figure 1B).

### 3.3. Clonality of BRAF Variant Classes

Overall, *BRAF^V600E^* mutations were more likely to be clonal (70.1%), while *aBRAF* mutations (class II, III, and unclassified *BRAF* variants) were more likely to be subclonal (62.6%, 56.0%, and 78.8%, respectively) (Table 2, Figure 2). While clonal V600 variants were found more commonly in female patients (60.2%), subclonal *BRAF* variants were more common in male patients (54.2–67.5%), regardless of classes. The median variant allele frequencies (VAFs) of clonal variants were 6.3% in V600 mutations, 7.4% in class II, 8.1% in class III, and 2.7% in unclassified variants (Table 3). The VAFs of subclonal variants ranged from 0.2% to 0.3% in the four classes. The cohort with subclonal *BRAF^V600E^* mutations presented with older patients (≥65 years, 52.8%), while the cohorts with subclonal *BRAF* non-V600 alterations had younger patients (<65 years, 59.1–65.3%),

### 3.4. Clonality and Anti-EGFR Exposure Score

Due to the lack of treatment history data in the GH cohort, a previously validated scoring system was applied to predict the exposure to anti-EGFR therapy. Patients were divided into two groups: those with predicted prior exposure (n = 3470) and those without predicted prior exposure (n = 11,272). Among patients with *BRAF* class II/III/unclassified variants, the proportion of patients with predicted prior exposure was more than two-fold greater than that in patients predicted to be non-exposed (2.1% vs. 0.8% in class II, 3.7% vs. 1.4% in class III, 6% vs. 2.6% in unclassified, Figure 3).

The *BRAF*-mutated patients were further grouped into those with clonal *BRAF* variants and those with subclonal *BRAF* variants. Most *BRAF^V600E^* mutations in patients with no predicted anti-EGFR exposure were clonal (Table 4, Figure 3). On the contrary, most *aBRAF* variants in patients with predicted prior exposure were subclonal (Table 4, Figure 3).

### 3.5. Co-Mutations Analysis

Co-mutation analysis was conducted for *BRAF*, *KRAS*, *NRAS*, *NF1*, *ERBB2*, *PIK3CA,* and *SMAD4* genes after excluding amplifications and synonymous alterations (Figure 4A,B). *aBRAF* variants often co-occurred with *KRAS* mutations, although more frequently in patients with prior anti-EGFR exposure (Figure 4A). Co-occurring *KRAS G12C* was only noted in one patient with a BRAF class II variant. Concomitant *NRAS* mutations were seen in 26.9–42.2% of patients with *aBRAF* variants and predicted the prior anti-EGFR exposure (Figure 4A) and were observed in only 2.7–5.8% of patients without predicted prior anti-EGFR exposure (Figure 4B). Co-mutations in *BRAF* and four other genes (*NF1*, *ERBB2*, *PIK3CA,* and *SMAD4*) were also more frequent in patients with predicted anti-EGFR exposure (Figure 4A); however, there was no significant difference between the *BRAF^V600E^* and *aBRAF* variant groups.

Blood-based tumor mutation burden (bTMB) values were available in 258 patients with *BRAF* variants. The median bTMB values in patients bearing *BRAF^V600E^*, class II, III, and unclassified variants were 12.44, 15.79, 12.44, and 31.58 mut/MB, respectively (Table 5, Figure 5).

### 3.6. Clinical Cohort Analysis

Our MD Anderson clinical cohort included 38 patients with *aBRAF* mCRC. The cohort’s median age was 55, 81% patients were Caucasian, and 74% had left-sided primary tumors (45% rectal, 24% sigmoid), with 37% being exposed to at least two lines of therapy (Appendix A). The most common *aBRAF* mutation was D594G (class III). The median follow-up time was 23.8 months (mo). The most common site of metastases involved was the liver. While there were no differences in OS between *aBRAF* classes, there was a significant difference in OS in patients with *RAS* co-mutations (28.3 mo, *p* = 0.05, Figure 6A) or liver involvement (28.8 mo, *p* = 0.02, Figure 6B). Patients < 50 years of age had extremely poor survival with an OS of 16.3 mo and an HR of 7.51 (95% CI = 1.82–31.01, *p* = 0.005, Figure 6C). Treatment with anti-EGFR (Figure 6D) or the use of metastasectomy (Figure 6E) did not result in statistically significant improvement in survival.

## 4. Discussion

Our study supports that *aBRAF* mCRC is indeed a distinct subtype of colorectal cancer with the most notable findings of ctDNA analysis, confirming that *aBRAF* (BRAF class II, III, unclassified variants) and co-occurrence with *KRAS*/*NRAS* mutations was increased at baseline and especially in patients predicted to have prior anti-EGFR exposure, in contrast to *BRAF^V600E^*. *RAS* co-mutation status was not previously consistently reported for *aBRAF* and highlights a unique molecular phenotype. We hypothesized this molecular phenotype would translate into a more aggressive clinical course, which was supported by the results of our MD Anderson clinical cohort. We identified *aBRAF* mCRC to be more common in younger patients with a median age of 55, more likely to have left-sided (sigmoid/rectal) primaries and liver metastases with worse outcomes for co-mutation with *RAS* and young onset. Anti-EGFR exposure and metastasectomy did not statistically improve survival outcomes in this subset. These findings reiterate the current void in treatment options for *aBRAF* mCRC and supports the need for novel therapeutic development.

As of 2023, there are no specific guidelines for the management of patients with *aBRAF* mCRC, with early-phase clinical trials generally recommended after the failure of traditional systemic 5-FU-based chemotherapy. Additionally, the efficacy of EGFR receptor antibodies remains inconclusive with data, suggesting that response may in fact be driven by the underlying *aBRAF* class [17,22,23,24,25]. Previous reports noted that class III *aBRAF* could benefit from anti-EGFR exposure; however, a subsequent study highlighted a lack of durable response and suggested class II *aBRAF* as a negative predictive biomarker for anti-EGFR receptor therapy [22,23,24,25]. In the present study, we showed significant differences between *BRAF^V600E^* and class II/III/unclassified variants in terms of prevalence and clonality, following the previous anti-EGFR treatment. The most common *aBRAF* variants that are usually encountered in clinical practice are class II G469A and class III D594G [22], which are consistent with our ctDNA findings. It should be noted that the present study showed a notable prevalence for unclassified variants, whose biology is unclear. Therefore, further investigation into the biology of these alterations is warranted. Our results show that both class II and III *aBRAF* mutations are associated with concomitant *RAS* aberrations at a frequency higher than that reported previously [18,19,25]. Prior anti-EGFR exposure significantly increased sub-clonal class II, III, and unclassified *aBRAF* mutation frequencies. This finding supports the hypothesis that *aBRAF* mutations may represent a resistance mechanism following EGFR inhibition in CRC [22].

In the current series, our MD Anderson cohort suggested EGFR inhibition in combination with chemotherapy may have limited efficacy in *aBRAF* mCRC. Although ongoing studies will further investigate this prospectively, such as the EPOC1703 study where both *aBRAF* class III patients who are EGFR receptor antibody naïve or refractory will be included and treated with the BEACON regimen [26], our data reveal that a significant proportion of patients with *aBRAF* have concomitant *RAS* mutations, thus limiting their eligibility and subsequent response to such a targeted approach. The data from our clinical cohort show that patients with double mutations in *aBRAF* and *RAS*, noted in both class II and class III *aBRAF*, have in fact inferior OS when compared to RAS wild-type (wt) *aBRAF* patients. Our results suggest that this double mutation phenotype represents in fact a more aggressive subset of mCRC, with a current void in the available clinical trials. Considering the high frequency of co-mutations and the unfavorable prognosis shown by double-mutated patients, novel approaches are therefore urgently needed. Interestingly, the analysis of bTMB data revealed that patients affected by all considered *BRAF* variants presented a median TMB value higher than 10 mut/MB, which may have implications for exposure to immune checkpoint inhibitors (ICI) therapy. However, these findings need to be interpreted with caution as blood-based TMB is typically higher than tissue-based TMB, with 16 mut/MB in blood roughly correlating to 10 mut/MB in tissue. Following the results of the KEYNOTE 158 trial [27], in 2020, the FDA approved anti-PD-1 inhibitors for any type of solid tumors with TMB ≥ 10 with tissue NGS. Our findings may be particularly useful for patients with unclassified *aBRAF* variants, since there are currently no specific therapeutic guidelines for this subtype, and they present the highest median bTMB value (31.58). A recent study suggested 28 mut/MB as a potential cutoff value to determine when ICI therapy may be more likely to be beneficial in CRC via a blood-based TMB measurement, but this needs to be corroborated by further research [28].

Several clinical trials targeting *aBRAF* mutations are currently ongoing. One therapeutic strategy attempted in these trials is to inhibit the MAPK pathway with novel MEK or ERK inhibitors, alone or combined with RAF inhibitors [29,30,31]. The examples of studies pursuing this strategy are NCT02465060, NCT02607813, NCT04249843, NCT02428712, and NCT03839342 [31]. Another strategy currently under investigation in early-phase clinical trials utilizes a novel Src homology phosphatase 2 (SHP2) inhibitor in cancers with class III mutations (NCT04045496; NCT03518554). SHP2 is a major scaffold protein downstream of numerous receptor tyrosine kinases, promoting RAS/MAPK signaling in cancers with class III *BRAF* mutations with concomitant *RAS* mutations. We have shown that the survival for these patients is much worse than for those presenting *aBRAF* RAS wt in our MD Anderson cohort. Unfortunately, none of these patients in this cohort had a concomitant *KRAS G12C* mutation. Therefore, a combination approach with a novel RAF dimer and G12C inhibitor would not be a feasible immediate path forward, while pan RAS inhibitors or checkpoint inhibition may be considered as viable options in the future.

This study presents some limitations. First, this is a retrospective, non-randomized cohort analysis. Second, because of the rarity of atypical *aBRAF* mutations, only 38 patients were identified for inclusion in our MD Anderson clinical cohort. Third, we were not able to verify if the patients from the ctDNA cohort underwent anti-EGFR therapy as treatment information is not available. Therefore, we applied a previously validated method that allowed to predict if patients were previously treated with anti-EGFR drugs.

## 5. Conclusions

In conclusion, we have highlighted clear differences in clonality between patients with atypical, non-V600 mutations and with traditional *BRAF^V600E^* mutations. We also summarized key aspects of *aBRAF* mutations as potential resistance mechanisms in *RAS/RAF* wt patients treated with anti-EGFR therapy from our ctDNA cohort. These results may help inform future anti-EGFR re-challenge strategies in future clinical trial design. Additionally, for the first time, we report on the co-mutation status as being a characteristic in both class II and class III atypical mutations based on our ctDNA cohort and highlight this feature representing an aggressive subtype of double-mutated mCRC (*RAS* & *aBRAF*) as a particularly difficult-to-treat patient population validated in our MD Anderson clinical cohort. Furthermore, it is important to highlight the need for additional research efforts regarding the preclinical characterization of unclassified variants. The unclassified cohort represents a group of patients whose underlying signaling biology is unclear to date, and such information would be informative for treatment decisions and early phase clinical trial triage regarding novel target/agent selection. Anti-EGFR exposure and metastasectomy do not appear to statistically impact survival outcomes in this subset, albeit we had a small cohort available to perform this analysis. However, all of these insights coupled together highlight the critical need for innovative clinical trials that consider these molecular and clinical intricacies. Additional innovative targeted approaches for *aBRAF* mCRC that address the co-mutation status may provide a viable path forward in this aggressive subset of colorectal cancer. These data represent the foundational framework for understanding the intricacies of *aBRAF* mCRC and highlight the need for continued dedicated therapeutic development for these unique patients.

## Figures and Tables

**Figure 1 cancers-16-00737-f001:**
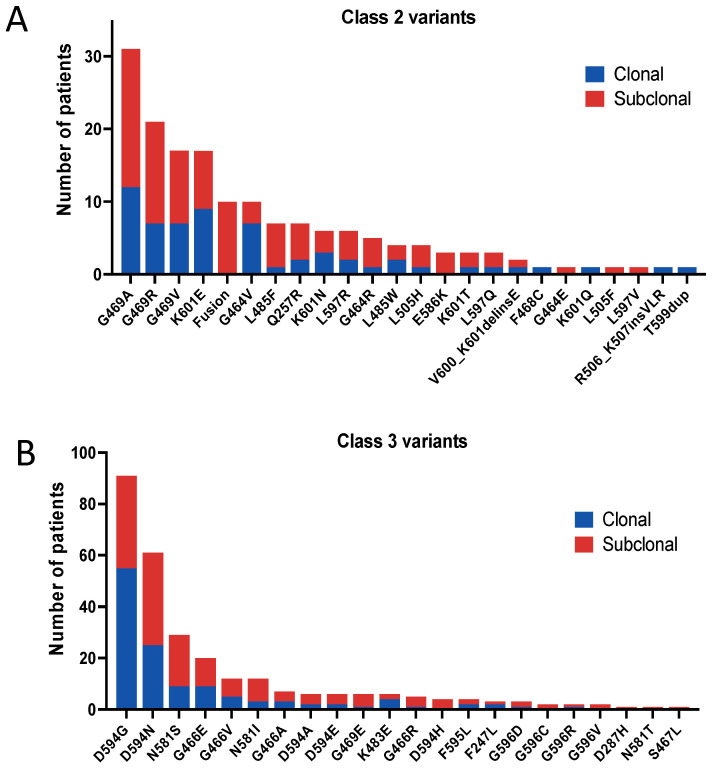
Number of patients expressing class II (**A**) and class III (**B**) variants. Clonal variants are presented in blue and subclonal variants in red.

**Figure 2 cancers-16-00737-f002:**
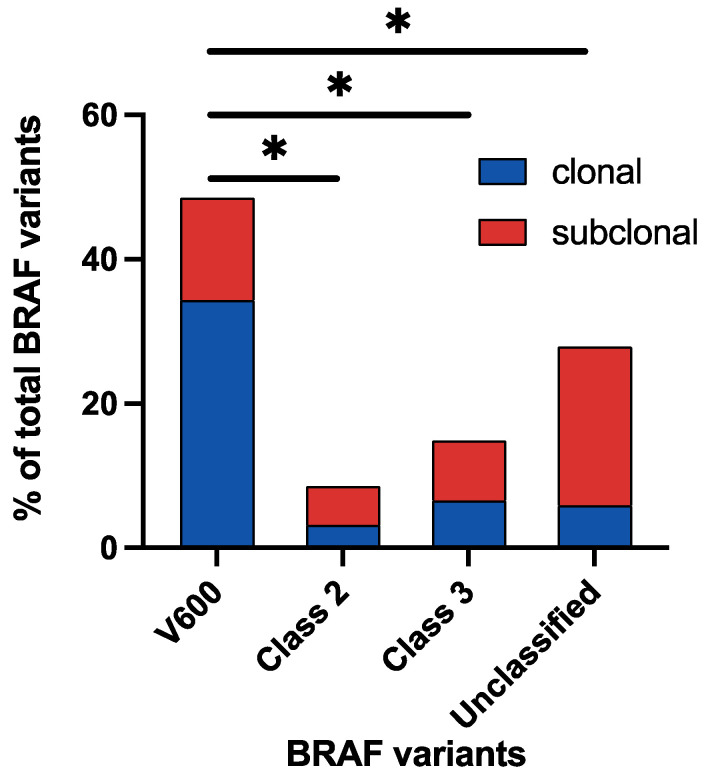
Clonality of BRAF variants in the ctDNA cohort. *: *p* < 0.0001.

**Figure 3 cancers-16-00737-f003:**
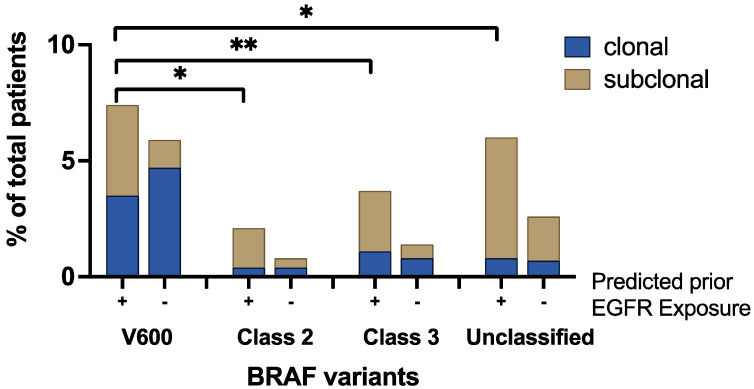
Clonality of BRAF variants in presence or absence of predicted anti-EGFR exposure. *: *p* < 0.0001; **: *p* = 0.001.

**Figure 4 cancers-16-00737-f004:**
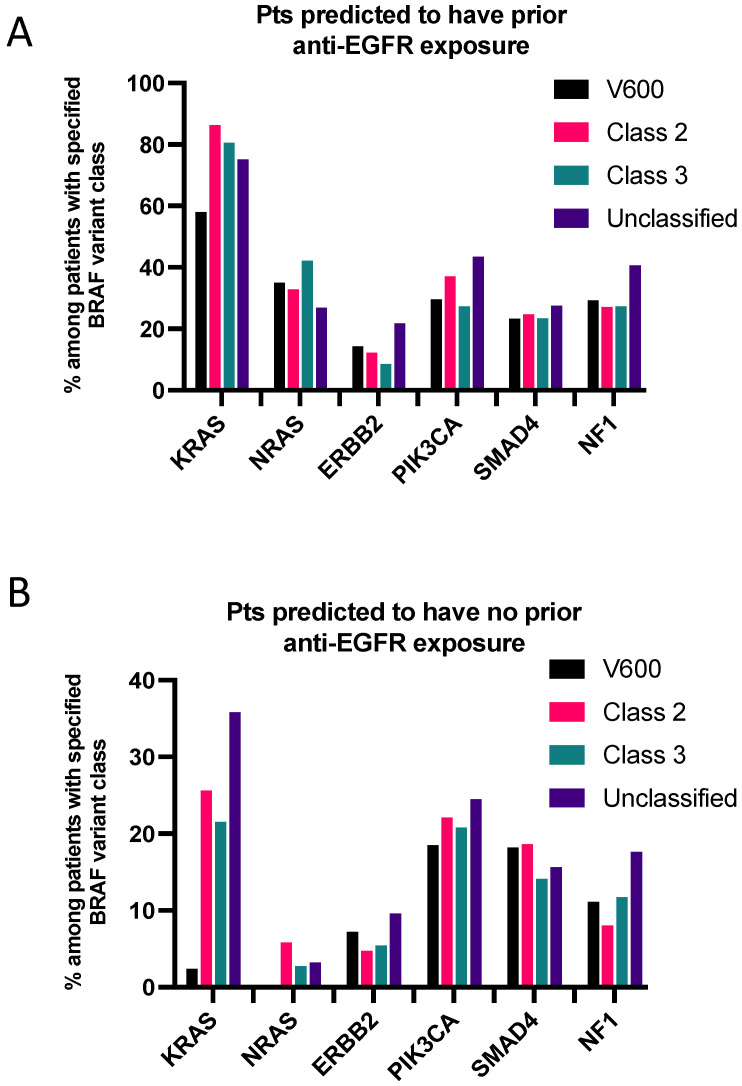
Frequency of co-mutations in patients (pts) predicted to have (**A**) or not have (**B**) prior exposure to anti-EGFR therapy.

**Figure 5 cancers-16-00737-f005:**
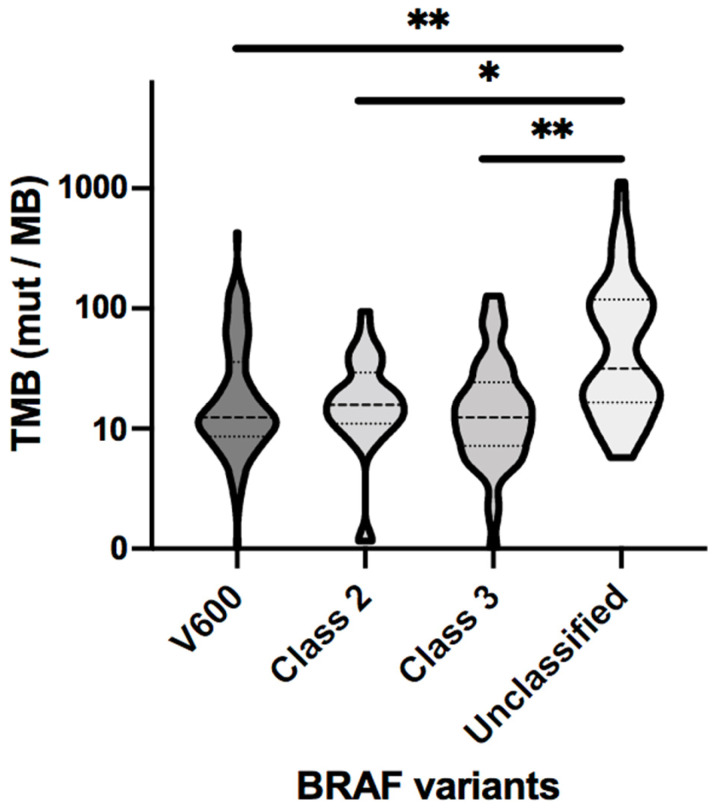
Violin distribution graphs of bTMB values for the different *BRAF* mutation classes. * *p* < 0.001; ** *p* < 0.0001.

**Figure 6 cancers-16-00737-f006:**
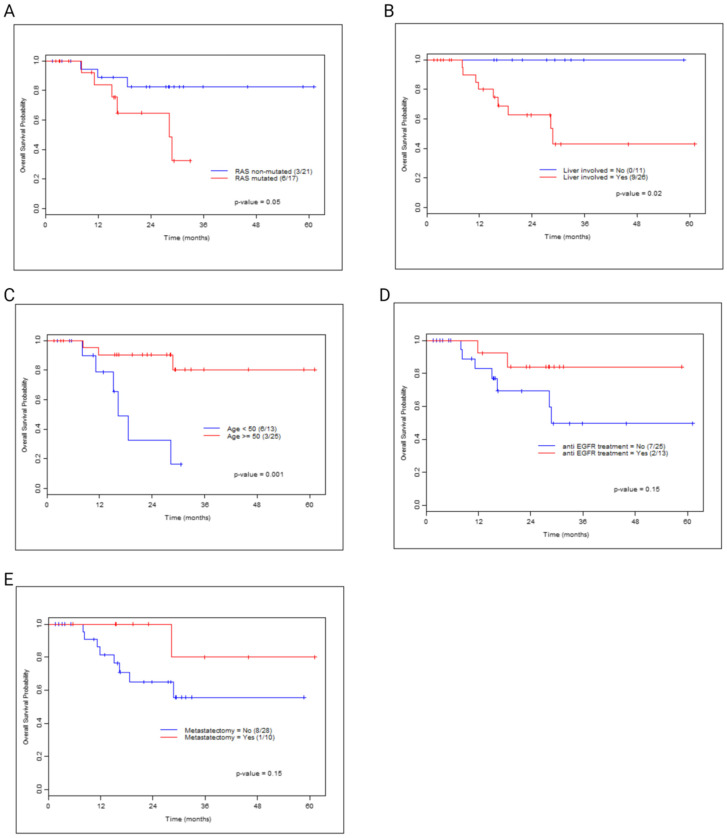
Kaplan–Meier survival curves comparing *aBRAF* patients in the clinical cohort based on *RAS* co-mutations (**A**), liver involvement (**B**), and the age at diagnosis, higher or lower than 50 (**C**), a history of anti-EGFR treatment (**D**), and a history of metastasectomy (**E**).

**Table 1 cancers-16-00737-t001:** Summary of patient characteristics from the ctDNA database/GH cohort.

BRAF Variants	14,742 mCRC Patients, 1733 Patients with BRAF Variants 1905 Total Variants, 431 Unique Variants			
	V600	Class II	Class III	Unclassified
Pts (% of BRAF pts, % of total CRC pts)	926 (53.4%, 6.3%)	159 (9.1%, 1.1%)	277 (16.0%, 1.9%)	475 (27.4%, 3.2%)
Variants (% of total variants)	926 (48.6%)	163 (8.6%)	284 (14.9%)	532 (27.9%)
Gender				
Male	408 (44.1%)	94 (59.1%)	165 (60.0%)	267 (56.2%)
Female	518 (55.9%)	65 (40.9%)	112 (40.4%)	208 (43.8%)
Age, years, median (range)	65 (16–98)	61 (28–95)	59 (28–94)	61 (14–95)
≥65	472 (51.0%)	59 (37.1%)	113 (40.8%)	190 (40.0%)
<65	451 (48.7%)	100 (62.9%)	163 (58.8%)	284 (59.8%)
NA	3 (0.3%)	0 (0%)	1 (0.4%)	1 (0.2%)

**Table 2 cancers-16-00737-t002:** Clonality of *BRAF* variants in the ctDNA cohort.

% of Total *BRAF* Variants (% of the Class)	Clonal Variant	Subclonal Variant
V600	34.3% (70.7%)	14.2% (29.3%)
Class II	3.2% (37.4%)	5.4% (62.6%)
Class III	6.6% (44.0%)	8.3% (56.0%)
Unclassified	5.9% (21.2%)	22% (78.8%)

**Table 3 cancers-16-00737-t003:** VAF values for clonal or subclonal variants of the different BRAF classes.

	Clonal Median VAF (Range)	Subclonal Median VAF (Range)
V600	6.3% (0.03–94.9%)	0.2% (0.01–36.0%)
Class II	7.4% (0.05–75.5%)	0.2% (0.03–14.7%)
Class III	8.1% (0.05–55.7%)	0.2% (0.01–27.6%)
Unclassified	2.7% (0.10–55.1%)	0.3% (0.04–31.6%)

**Table 4 cancers-16-00737-t004:** Clonality of *BRAF* variants in presence or absence of predicted anti-EGFR exposure.

	V600	Class II	Class III	Unclassified
EGFR exposure				
clonal	3.5%	0.4%	1.1%	0.8%
subclonal	3.9%	1.7%	2.6%	5.2%
no EGFR exposure				
clonal	4.7%	0.4%	0.8%	0.7%
subclonal	1.2%	0.4%	0.6%	1.9%

**Table 5 cancers-16-00737-t005:** Sample size and bTMB median values for the different *BRAF* classes.

	Number of Samples	TMB Median (mut/MB)
V600	120	12.44
Class II	36	15.79
Class III	45	12.44
Unclassified	57	31.58

## Data Availability

Deidentified data are available if requested after approval from all authors and Guardant Health.

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
