# Peer review of "Comprehensive Landscape of BRAF Variant Classes, Clonalities, and Co-Mutations in Metastatic Colorectal Cancer Using ctDNA Profiling"

_cancers, 2024, doi:10.3390/cancers16040737_

Round 1

Reviewer 1 Report

Comments and Suggestions for Authors

Major:

1. In line 46, the authors state "BRAFv660e and RAS mutations are mutually exclusive."  However, figure 4 shows  BRAFv600-RAS co-mutations and in line 173, the BRAFv600 mutations shown in Fig. 4 are named BRAFv600e. How to reconcile the literature cited in the introduction and the findings presented. This should be discussed, or - in case this reviewer misunderstood/did not get the point to be made - explained much more clearly and explicitly.

2. Material & methods: The authors refer to previous studies as data sources. However, they should provide more detailed information on the method/procedure of the "tissue next generation sequencing" performed on samples from the clinical cohort (38 patients) at MD Anderson Center. How were the biopsies taken and how was the material processed?

Minor: 

- ll. 252-255: "However, these findings need to be interpreted with caution as blood based TMB typically trends higher than tissue based TMB, with 16mut/MB on tissue roughly correlating to 10mut/MB in blood." Is this statement correct? Or should the numbers (16 and 10) be interchanged?

- ll. 285-288: "We also summarized key aspects of aBRAF mutations as potential resistance mechanisms in RAS/RAF wt patients treated with anti-EGFR therapy from our ctDNA cohort and should be considered in future anti-EGFR re-challenge strategies." Syntax of this sentence? Especially the second part "and should be considered....".   

Typos/Errors:

- line 104: "..was used to compared PS between groups." Remove "d" from compared. 

- line 180: first 12,44. Comma should be replaced by dot.

- line 223: "...could benefit from EGFR exposure...". Was it meant to read "...could benefit from anti-EGFR exposure..." or "...EGF-exposure...."?

- same line: replace "semicolon" between "exposure" and "However" by a colon/dot.

- line 304 : "Data availability statement: De identified data", "De" should be "The". 

Author Response

Responses to Reviewer #1:

Major:

  1. In line 46, the authors state "BRAFv660e and RAS mutations are mutually exclusive."  However, figure 4 shows  BRAFv600-RAS co-mutations and in line 173, the BRAFv600 mutations shown in Fig. 4 are named BRAFv600e. How to reconcile the literature cited in the introduction and the findings presented. This should be discussed, or - in case this reviewer misunderstood/did not get the point to be made - explained much more clearly and explicitly. Response: In the literature and in clinical practice BRAFV600E is predominantly mutually exclusive from RAS driver mutations, however it is not absolutely zero percent incidence.  I did add the word “predominantly” to cover this point.  In regards to Figure 4 there is sub group A and sub group B.  Figure 4A represents patients who previously received an anti-EGFR drug and we know these patients develop resistance mechanisms, one of which is increased KRAS and NRAS mutations. However when you look at Figure 4B (the group of patients who did not receive prior anti-EGFR drug) the incidence of KRAS is very low and NRAS is zero. 
  2. Material & methods: The authors refer to previous studies as data sources. However, they should provide more detailed information on the method/procedure of the "tissue next generation sequencing" performed on samples from the clinical cohort (38 patients) at MD Anderson Center. How were the biopsies taken and how was the material processed? Yes, I added text to describe NGS at the cancer center

Minor: 

- ll. 252-255: "However, these findings need to be interpreted with caution as blood based TMB typically trends higher than tissue based TMB, with 16mut/MB on tissue roughly correlating to 10mut/MB in blood." Is this statement correct? Or should the numbers (16 and 10) be interchanged? Yes, This statement is corrected. 

- ll. 285-288: "We also summarized key aspects of aBRAF mutations as potential resistance mechanisms in RAS/RAF wt patients treated with anti-EGFR therapy from our ctDNA cohort and should be considered in future anti-EGFR re-challenge strategies." Syntax of this sentence? Especially the second part "and should be considered....".   Addressed and updated. 

Typos/Errors:

- line 104: "..was used to compared PS between groups." Remove "d" from compared. -corrected

- line 180: first 12,44. Comma should be replaced by dot.- corrected

- line 223: "...could benefit from EGFR exposure...". Was it meant to read "...could benefit from anti-EGFR exposure..." or "...EGF-exposure...."? -corrected

- same line: replace "semicolon" between "exposure" and "However" by a colon/dot. -corrected

- line 304 : "Data availability statement: De identified data", "De" should be "The". -corrected to not have a space should be deidentified data, not “identified”. 

Reviewer 2 Report

Comments and Suggestions for Authors

This study describes the analysis of BRAF mutations in metastatic colorectal cancer using ctDNA. Findings include differences in clonality between BRAFV600E and atypical BRAF mutations. Clinical implications of the different groups are discussed.

A question that hasn’t been touched on is ‘How are patients treated that have both a BRAFV600E mutation and an additional atypical BRAF mutation’?

Introduction

-       It is unsure how well ctDNA reflects the genomic profile of mCRC. Without analysing the actual tumour genomes and their mutation profiles, it might be a bit ambitious to state that this study using ctDNA can provide a comprehensive landscape of BRAF mutations in mCRC (line 72+). Maybe the authors could limit this statement to ctDNA or provide references that show that the ctDNA sufficiently represents the genomic landscape of the tumour.

Methods

-          Section 2.1 - How many patients were in the first cohort? What material was analysed with the Guardant360 targeted sequencing assay? Is the second clinical cohort (n=38) a subset of the first large cohort or independent and separate?

-          aBRAF variants in the first GH cohort can be class II/III/unclassified, is that the same for the MD Anderson cohort or are these specifically class II/III variants? If so please add these details here.

-          Please provide more details about the next generation sequencing used to validate aBRAF variants

-          Please provide more details on when the blood and ctDNA was extracted and how it was sequenced. What was the sequencing depth? Without that information it’s hard to know if it was sufficient to pick up variants with low MAF. What was the ctDNA concentration? In other studies this has been used as biomarkers so would be good to have that information here as well. What does ‘baseline’ mean in this setting? Prior to any treatment? Does that mean the ctDNA for the other 28 patients was taken after treatment?

-          Line 91 ‘pts’ has not been introduced as an abbreviation (I assume for patients)

-          Section 2.2 line 97 - is this an established cut-off to define clonality for ctDNA variants using MAF?

-          Section 2.2 line 103 - The authors defined OS as time from mCRC to date of death/last follow up which seems appropriate. However, the part about ‘Kaplan-Meier method was used to estimate OS” in the next sentence is confusing unless there are 2 different OS used in the study? It could be rephrased to something like ‘Kaplan-Meier survival curves and long-rank tests were used to compare OS between patient groups.’

-          How were patients handled that had multiple BRAF variants that were in different classes (e.g. a V600 and a class II aBRAF variant)? Should these patients be excluded from the analyses since they don’t fully fit in either class or could they form their own group of a ‘mixed BRAF status’?

Results

-          Section 3.1 - In the text 284 patients have a class III BRAF variant, however table 1 states 277. The table also adds up to a total of 1837 patients with a BRAF variant, when the text states 1733. Please ensure consistency between text and tables. Is the ctDNA database (table 1 legend) the GH cohort (text)?

-          Table 1 - Would it be possible to add the predicted anti-EGFR exposure status to the table for each of the patient groups to see the differences in (predicted) treatment history between the groups?

-          Section 3.2 - Figure 1 includes colouring by clonal status of the clonal status of the variants found in each codon. However, there is no mention of it in the text. Are there any observations that the readers should pay attention to?

-          Can the explanation of ‘aBRAF mutations (class II, III and unclassified BRAF variants)’ be moved from line 131+ to line 115?

-          Is it noteworthy that subclonal BRAFV600 patients are older than subclonal aBRAF patients, since that has been mentioned in the overall cohort description of the 2 groups (lines 114) and no data is shown for it?

-          Figure 2 – Can this figure be incorporated into Figure 1 to show the overall distribution of variants between the classes? In its current form it doesn’t add much since it mainly visualises the 4th row of table 1. This section is about the clonality of BRAF variants so it would be more interesting to see if there are statistical differences between the number/percent of clonal V600 and aBRAF variants and between the number/percent of subclonal V600 and aBRAF variants. At the moment it shows that that are significantly more v600 variants than aBRAF variants which fits better into the previous section.

-          Table 4 - The percentages in this table don’t add up to 1, however all BRAF variants should be in one of the groups.

-          Figure 3 - what differences are the p-values assessing? The percentage of patients with EGFR exposure stratified by their BRAF variants (V600 vs the other aBRAF classes) or does it take the clonality into account as well? Please provide more information in the figure legend on which comparisons were performed to generate those p-values (similar in Figure 2).

-          Section 3.5 – the previous sections split the BRAF mutations into clonal/subclonal populations. Would it be possible to do this here as well? Is it more likely to have co-occuring mutations if the BRAF variant is clonal?

-          Figure 5 - in comparison to the other figures this figure feels unnecessarily zoomed in with massive font sizes (way larger than the text surrounding it). Table 5 and figure 5 could be combined to just figure 5. The number of samples could be added as n=x at the x-axis labels and the medians are mentioned in the text and indicated in the figure. Please change TMB to bTMB as introduced in the text and add that abbreviation to the figure legend.

-          Section 3.6 - Are these 38 patients a subgroup of the initially described cohort? Supplementary Table 1 would be good to be a main table since the clinical information for this cohort are of interest. It would also be good to know how many of these patients had class II/III/unclassified BRAF variants. Depending on numbers in each group it might not be possible to draw meaningful conclusions. Please include median OS/follow-up times.

-          Please provide a Kaplan-Meier plot to show the survival curves for the different aBRAF classes.

-          Line 193 - Please provide a Supplementary table with the aBRAF variants (with added RAS mtuations) that can be referred to here.

-          Lines 1965+ - Please provide details on which patient group has improved/worse survival instead of stating that there was a significant survival difference.

-          For the first cohort the age cut-off was 65, here it is 50. Could this be made consistent so that the cohorts can be compared better for this characteristic and to have a consistent threshold of young/old patients for this disease.

-          Lines 198+, 214+, 292+ - In Figure 6D/E, patients with anti-EGFR/metastasectomy showed better survival than patients not treated with anit-EGFR/metastasectomy which contradicts the description in the text. Please double check the figure and text to ensure that all labelling and descriptions are correct. This might also further impact the discussion (lines 237+)

-          I am a bit confused about the differences between figures 2 and 3 in regard to clonal/subclonal proportions in the different classes. In figure 2, approximately half of the class III variants are shown to be subclonal, however in Figure 3 when the same variants are split into EGFR exposure +/- there appear to be significantly more subclonal than clonal variants in the class III (when stacking the two bars for this group together). Similarly for the class II variants.

Discussion

-          Line 206+ - Previously aBRAF variants were class II/III and unclassified. Do the authors intentionally refer to only a subset of aBRAF variants here?

-          Line 207+ - From the described results the reference to baseline is unclear since no details about when the ctDNA samples were taken or if a timecourse experiment was conducted to come to this conclusion.

-          Line 226 - What expression are the authors referring to here? The study does not include any gene expression data or references to it.

-          Line 272 - Is there a ‘RAS’ missing before the wt? aBRAF wt does not make sense here since the full Anderson cohort supposedly had aBRAF mutations.

General comments

-          There seem to be double-spaces in lines 40, 51, 82, 90, 93, 193, 194, 197 and more at the beginning of new sentences.

-          Table 1 - Please introduce ‘pts’ as the abbreviation for patients in the table legend

-          If MAF (methods) and VAF (results) are used interchangeably, please pick one for consistency

-          Figure 6 - Please increase the font size within the Kaplan-Meier plots and add n=X to each of the groups. What do the current values in brackets mean (e.g 3/21 in A)? Since blue and red have previously been used to separate clonal and subclonal BRAF variants it might be better to use different colours for the Kaplan-Meier plots (or leave the colours for the Kaplan-Meier plots and change the clonal/subclonal colours).

-          Please use consistent terms. ‘BRAF non-V600’ as well as ‘BRAF class II/III/unclassified’ have been introduced as aBRAF variants so this term should be used throughout the manuscript for easy recognition and readability (line 140 and 150)

-          Figure 1, 2, 3 - Could the authors change the ‘Class 2’ to ‘Class II’ and ‘Class 3’ to ‘Class III’ in the figures for consistency?

-          Figure 3 - please use consistent colours for the same information (clonal = blue, subclonal = red in previous figures)

-          The figure and table legends are very brief and would benefit from more details especially if p-values are indicated it would be good which groups were compared when additional colouring is provided to split the bars.

-          Line 165+ - since Figure 4 only has parts A and B, it would be enough to refer to ‘Figure 4’ here instead of Figure 4A-B.

-          Line 180 - the decimal character should be a ‘.’ for ’12.44’

-          The results could benefit from more connections between the section as to why the authors decided to look at each of the analyses. Quite often the new paragraph starts a bit out of the blue and doesn’t feel like a cohesive flow (e.g. the bTMB).

-          Line 198 - OS between the age groups refers to ‘Figure 6C’ not ‘Figure 6B’

-          Inconsistent spelling (e.g. subclonal or sub-clonal)

-          Line 236 CRC or mCRC?

-          Line 251 - TMB or bTMB?

-          Line 278 - aBRAF or BRAF?

-          Line 287 - what are RAS/RAF wt patients?

-          Line 292 - ‘MDA’ has not been introduced as an abbreviation

Author Response

Responses to Reviewer #2:

Introduction

-       It is unsure how well ctDNA reflects the genomic profile of mCRC. Without analysing the actual tumour genomes and their mutation profiles, it might be a bit ambitious to state that this study using ctDNA can provide a comprehensive landscape of BRAF mutations in mCRC (line 72+). Maybe the authors could limit this statement to ctDNA or provide references that show that the ctDNA sufficiently represents the genomic landscape of the tumour. – acknowledged and modified the sentence. 

Methods

-          Section 2.1 - How many patients were in the first cohort? What material was analysed with the Guardant360 targeted sequencing assay? Is the second clinical cohort (n=38) a subset of the first large cohort or independent and separate? -  in the guardant cohort 1,733 out of 14,742 mCRC patients in the ctDNA cohort had at least one BRAF variant, as listed and described in section 3 and Table 1.  The guardant 360 assay is published and commercially available, but essentially a liquid biopsy assay with coverage as described in the text.   The second clinical cohort is a separate set of patients identified to have atypical BRAF treated at MD Anderson Cancer Center.  The main purpose of this clinical cohort was to look at the impact of co mutation status with KRAS or NRAS with atypical BRAF in terms of outcomes and confirm our hypothesis that co mutation would result in worse outcomes. 

-          aBRAF variants in the first GH cohort can be class II/III/unclassified, is that the same for the MD Anderson cohort or are these specifically class II/III variants? If so please add these details here.   – The MD Anderson cohort are atypical BRAF mutations picked up on tissue based next generation sequencing and treated at our cancer center.  This could have included unclassified, but the majority were class II or III.   

-          Please provide more details about the next generation sequencing used to validate aBRAF variants – added in the text

-          Please provide more details on when the blood and ctDNA was extracted and how it was sequenced. What was the sequencing depth? Without that information it’s hard to know if it was sufficient to pick up variants with low MAF. What was the ctDNA concentration? In other studies this has been used as biomarkers so would be good to have that information here as well. What does ‘baseline’ mean in this setting? Prior to any treatment? Does that mean the ctDNA for the other 28 patients was taken after treatment?  In the ctDNA cohort, the assay is the Guardant 360 assay which is a commercial assay and its coverage, ctDNA concentration are all available in the literature, not included here in the text.  Unfortunately the guardant health cohort does not have available treatment history so timing of when the ctDNA was drawn is not clear from the cohort.  The use of an anti-EGFR signature was applied from our prior publication to get a sense of exposure to certain targeted therapy, but the ctDNA analysis was primarily used to speak to co mutation status and clonality.  For the 38 patients at MD Anderson the atypical BRAF call was based on tissue NGS.  Baseline ctDNA here was referring to a confirmatory ctDNA obtained prior to treatment start at MD Anderson.  I can remove the baseline terminology as that is confusing. 

-          Line 91 ‘pts’ has not been introduced as an abbreviation (I assume for patients) - corrected

-          Section 2.2 line 97 - is this an established cut-off to define clonality for ctDNA variants using MAF? Per Guardant Health science leadership recommendations, our collaborator. 

-          Section 2.2 line 103 - The authors defined OS as time from mCRC to date of death/last follow up which seems appropriate. However, the part about ‘Kaplan-Meier method was used to estimate OS” in the next sentence is confusing unless there are 2 different OS used in the study? It could be rephrased to something like ‘Kaplan-Meier survival curves and long-rank tests were used to compare OS between patient groups.’ – acknowledged and corrected

-          How were patients handled that had multiple BRAF variants that were in different classes (e.g. a V600 and a class II aBRAF variant)? Should these patients be excluded from the analyses since they don’t fully fit in either class or could they form their own group of a ‘mixed BRAF status’?  Its not a common scenario to have clonal BRAF V600E and clonal atypical BRAF, but technically a clonal BRAFV600E could develop subclonal atypical BRAF mutations as likely resistance mechanisms when exposed to anti-EGFR (in combination with BRAF inhibition), probably worth a separate analysis to really report incidence.  Overall, we believe this to be low. 

Results

-          Section 3.1 - In the text 284 patients have a class III BRAF variant, however table 1 states 277. The table also adds up to a total of 1837 patients with a BRAF variant, when the text states 1733. Please ensure consistency between text and tables. Is the ctDNA database (table 1 legend) the GH cohort (text)? Thank you, corrected in text.  Yes it is the GH cohort- Added to the table legend.

-          Table 1 - Would it be possible to add the predicted anti-EGFR exposure status to the table for each of the patient groups to see the differences in (predicted) treatment history between the groups? – We tried to show this via Table 4 and Figure 3

-          Section 3.2 - Figure 1 includes colouring by clonal status of the clonal status of the variants found in each codon. However, there is no mention of it in the text. Are there any observations that the readers should pay attention to? The main observation of Figure 1 is to give the reader a sense of the GH cohort and differences noted in aBRAF in terms of clonality. 

-          Can the explanation of ‘aBRAF mutations (class II, III and unclassified BRAF variants)’ be moved from line 131+ to line 115? – prefer to keep the order as is

-          Is it noteworthy that subclonal BRAFV600 patients are older than subclonal aBRAF patients, since that has been mentioned in the overall cohort description of the 2 groups (lines 114) and no data is shown for it? – I think the subclonal mutations are more a factor of prior treatment details (which the GH cohort does not provide) but that is speculation. 

-          Figure 2 – Can this figure be incorporated into Figure 1 to show the overall distribution of variants between the classes? In its current form it doesn’t add much since it mainly visualises the 4th row of table 1. This section is about the clonality of BRAF variants so it would be more interesting to see if there are statistical differences between the number/percent of clonal V600 and aBRAF variants and between the number/percent of subclonal V600 and aBRAF variants. At the moment it shows that that are significantly more v600 variants than aBRAF variants which fits better into the previous section. – I prefer the current set up as we are focusing on BRAFV600E in Figure 2 and initially Figure 1 is mainly describing distribution of GH cohort and then moving to specific clonality in the Figure 2, Table 2. 

-          Table 4 - The percentages in this table don’t add up to 1, however all BRAF variants should be in one of the groups. This is because we are comparing EGFR exposure to non. 

-          Figure 3 - what differences are the p-values assessing? The percentage of patients with EGFR exposure stratified by their BRAF variants (V600 vs the other aBRAF classes) or does it take the clonality into account as well?  Yes for Figure 3 and Figure 2 is comparing the differences in clonality alone. Please provide more information in the figure legend on which comparisons were performed to generate those p-values (similar in Figure 2).  The figure is showing this

-          Section 3.5 – the previous sections split the BRAF mutations into clonal/subclonal populations. Would it be possible to do this here as well? Is it more likely to have co-occuring mutations if the BRAF variant is clonal? – I am not able to perform new analysis as I have moved to a new institution

-          Figure 5 - in comparison to the other figures this figure feels unnecessarily zoomed in with massive font sizes (way larger than the text surrounding it). Table 5 and figure 5 could be combined to just figure 5. The number of samples could be added as n=x at the x-axis labels and the medians are mentioned in the text and indicated in the figure. Please change TMB to bTMB as introduced in the text and add that abbreviation to the figure legend. – can discuss with editor

-          Section 3.6 - Are these 38 patients a subgroup of the initially described cohort? Supplementary Table 1 would be good to be a main table since the clinical information for this cohort are of interest. It would also be good to know how many of these patients had class II/III/unclassified BRAF variants. Depending on numbers in each group it might not be possible to draw meaningful conclusions. Please include median OS/follow-up times.  Yes a subgroup and can discuss with editor about moving S1

-          Please provide a Kaplan-Meier plot to show the survival curves for the different aBRAF classes. Since it was not statistically significant we did not include here. 

-          Line 193 - Please provide a Supplementary table with the aBRAF variants (with added RAS mtuations) that can be referred to here.

-          Lines 1965+ - Please provide details on which patient group has improved/worse survival instead of stating that there was a significant survival difference. (RAS and liver involvement had worse survival)

-          For the first cohort the age cut-off was 65, here it is 50. Could this be made consistent so that the cohorts can be compared better for this characteristic and to have a consistent threshold of young/old patients for this disease. I do not have feasibility to change the figures at this state as I have moved from the insitution

-          Lines 198+, 214+, 292+ - In Figure 6D/E, patients with anti-EGFR/metastasectomy showed better survival than patients not treated with anit-EGFR/metastasectomy which contradicts the description in the text. Please double check the figure and text to ensure that all labelling and descriptions are correct. This might also further impact the discussion (lines 237+) – I was referring to statistical differences however I will add the numbers are small to make any firm conclusions. 

-          I am a bit confused about the differences between figures 2 and 3 in regard to clonal/subclonal proportions in the different classes. In figure 2, approximately half of the class III variants are shown to be subclonal, however in Figure 3 when the same variants are split into EGFR exposure +/- there appear to be significantly more subclonal than clonal variants in the class III (when stacking the two bars for this group together). Similarly for the class II variants. – main point here is figure 2 is BRAF class clonality distribution comparisons in GH cohort whereas point of figure 3 is to contrast with the various BRAF variant classes based on the presence or absence of the anti-EGFR score.

Discussion

-          Line 206+ - Previously aBRAF variants were class II/III and unclassified. Do the authors intentionally refer to only a subset of aBRAF variants here? (corrected thank you)

-          Line 207+ - From the described results the reference to baseline is unclear since no details about when the ctDNA samples were taken or if a timecourse experiment was conducted to come to this conclusion. (acknowledged and corrected)

-          Line 226 - What expression are the authors referring to here? The study does not include any gene expression data or references to it. corrected

-          Line 272 - Is there a ‘RAS’ missing before the wt? aBRAF wt does not make sense here since the full Anderson cohort supposedly had aBRAF mutations. - corrected

General comments

-          There seem to be double-spaces in lines 40, 51, 82, 90, 93, 193, 194, 197 and more at the beginning of new sentences. (editor to address)

-          Table 1 - Please introduce ‘pts’ as the abbreviation for patients in the table legend - spelled out

-          If MAF (methods) and VAF (results) are used interchangeably, please pick one for consistency – corrected to VAF

-          Figure 6 - Please increase the font size within the Kaplan-Meier plots and add n=X to each of the groups. What do the current values in brackets mean (e.g 3/21 in A)? Since blue and red have previously been used to separate clonal and subclonal BRAF variants it might be better to use different colours for the Kaplan-Meier plots (or leave the colours for the Kaplan-Meier plots and change the clonal/subclonal colours). – increased figure

-          Please use consistent terms. ‘BRAF non-V600’ as well as ‘BRAF class II/III/unclassified’ have been introduced as aBRAF variants so this term should be used throughout the manuscript for easy recognition and readability (line 140 and 150) – added to first mention in abstract

-          Figure 1, 2, 3 - Could the authors change the ‘Class 2’ to ‘Class II’ and ‘Class 3’ to ‘Class III’ in the figures for consistency? – thank you for the suggestion

-          Figure 3 - please use consistent colours for the same information (clonal = blue, subclonal = red in previous figures) – thank you for the suggestion

-          The figure and table legends are very brief and would benefit from more details especially if p-values are indicated it would be good which groups were compared when additional colouring is provided to split the bars. – thank you for the suggestion; however our GH collaborator prefers this style

-          Line 165+ - since Figure 4 only has parts A and B, it would be enough to refer to ‘Figure 4’ here instead of Figure 4A-B. can discuss with editor

-          Line 180 - the decimal character should be a ‘.’ for ’12.44’ - corrected

-          The results could benefit from more connections between the section as to why the authors decided to look at each of the analyses. Quite often the new paragraph starts a bit out of the blue and doesn’t feel like a cohesive flow (e.g. the bTMB). – address in conclusions

-          Line 198 - OS between the age groups refers to ‘Figure 6C’ not ‘Figure 6B’ - corrected

-          Inconsistent spelling (e.g. subclonal or sub-clonal) - noted

-          Line 236 CRC or mCRC? mCRC

-          Line 251 - TMB or bTMB? If blood based is written out I left it as TMB

-          Line 278 - aBRAF or BRAF? aBRAF

-          Line 287 - what are RAS/RAF wt patients? RAS/RAF wild type

-          Line 292 - ‘MDA’ has not been introduced as an abbreviation – will change

Reviewer 3 Report

Comments and Suggestions for Authors

The study presents the characterization and significance of atypical BRAF variants in metastatic colorectal cancer.

The manuscript is well written but needs clarification in the methods section:

- What type of samples were included in the first cohort? Pre-treatment ctDNA samples? Please add more details about the Guardant360 test, e.g. it is not mentioned in the text that it is a liquid biopsy test.

- What type of NGS testing was performed on the second cohort? Were these pre-treatment samples (tissue biopsies?) or otherwise? How were the samples (FFPE and blood) collected and processed? Please provide additional details.

Author Response

Responses to Reviewer #3:

The study presents the characterization and significance of atypical BRAF variants in metastatic colorectal cancer.

The manuscript is well written but needs clarification in the methods section:

- What type of samples were included in the first cohort? Pre-treatment ctDNA samples? Please add more details about the Guardant360 test, e.g. it is not mentioned in the text that it is a liquid biopsy test. – added text confirming this is a blood based, “liquid biopsy”.  Additionally, there is no treatment history available for the guardant database so we are unable to confirm the timeline of when the ctDNA was drawn.  However utilizing a anti-EGFR exposure signature we were able to predict (not confirm) whether the sample was a pre or post treatment for some cases. 

- What type of NGS testing was performed on the second cohort? Were these pre-treatment samples (tissue biopsies?) or otherwise? How were the samples (FFPE and blood) collected and processed? Please provide additional details. – added text to explain the NGS assay

Round 2

Reviewer 2 Report

Comments and Suggestions for Authors

Thank you to the author(s) to address my previous comments and concerns.

I have a few remaining points.

Figure 4. If the figure itself cannot be edited, please add the ‘Pts’ abbreviation to the figure legend.

Table 4. I still don’t understand why these percentages don’t add up to 1. This table should capture all BRAF mutations (100%) classified as before and split by EGFR exposure (yes/no) and clonal/subclonal.

Line 210+ - it is mentioned that there are no OS differences between aBRAF classes. Please provide the Kaplan-Meier plot that shows this since other non-significant plots are mentioned and shown here. Please provide details in the text on which patient group has improved/worse survival instead of stating that there was a significant survival difference

Regarding the different age cut-offs between the 2 cohorts and other figure related comments. Given all authors participated in all aspects of the project including the formal analysis, data curation and visualization, one of the co-authors might be able to edit the figures to improve the consistent flow and content of the manuscript.

Author Response

Thank you to the author(s) to address my previous comments and concerns.

I have a few remaining points.

- Figure 4. If the figure itself cannot be edited, please add the ‘Pts’ abbreviation to the figure legend.

Thank you.  Completed.

- Table 4. I still don’t understand why these percentages don’t add up to 1. This table should capture all BRAF mutations (100%) classified as before and split by EGFR exposure (yes/no) and clonal/subclonal.

Thank you.  We took the total GH cohort and divided into two groups: those with predicted prior exposure (n=3,470) and those without predicted prior exposure (n=11,272). The GH cohort consisted of 14,742 patients with mCRC, among whom 1,733 presented at least one BRAF variant.  Among patients with BRAF class II/III/unclassified variants, the proportion of patients with predicted prior exposure was more than two-fold greater than that in patients predicted to be non-exposed (2.1% vs 0.8% in class II, 3.7% vs 1.4% in class III, 6% vs 2.6% in unclassified, Figure 3.  This was generated by our Guardant colleagues and based on the description here. 

- Line 210+ - it is mentioned that there are no OS differences between aBRAF classes. Please provide the Kaplan-Meier plot that shows this since other non-significant plots are mentioned and shown here. Please provide details in the text on which patient group has improved/worse survival instead of stating that there was a significant survival difference

We previously have published and shown the difference of aBRAF as a whole (all classes) compared to BRAFV600E and wild type RAS in JCO PO 2019.  In this paper, we comment – “while there were no differences in OS between aBRAF classes, there was a significant difference in OS in patients with RAS co-mutations (28.3 mo, p=0.05, Figure 6A) or liver involvement (28.8 mo, p=0.02, Figure 6B)”.  We show what we believe to be the relevant finding.  I am unable to provide the OS curves and my colleagues are not able to do this for me with the limited time constraints and my inability to access data files with my new position. 

- Regarding the different age cut-offs between the 2 cohorts and other figure related comments. Given all authors participated in all aspects of the project including the formal analysis, data curation and visualization, one of the co-authors might be able to edit the figures to improve the consistent flow and content of the manuscript.

I am unable to provide any new analysis and change age cut offs for the data set considering my departure from the institution and other conflicts.  My co authors are also not able to do this new analysis with the time constraints and fluctuating responsibilities.  We have done our best to address these comments with this constraint.  Thank you for your understanding.